# Gut microbiota–derived short-chain fatty acids protect against the progression of endometriosis

Sangappa B Chadchan[1,2], Pooja Popli[1,2], Chandrasekhar R Ambati[5], Eric Tycksen[4], Sang Jun Han[5], Serdar E Bulun[6], Nagireddy Putluri[5], Scott W Biest[1,3], Ramakrishna Kommagani[1,2]

**Worldwide, ~196 million are afflicted with endometriosis, a painful disease in which endometrial tissue implants and proliferates on abdominal peritoneal surfaces. Theories on the origin of endometriosis remained inconclusive. Whereas up to 90% of women experience retrograde menstruation, only 10% develop endometriosis, suggesting that factors that alter peritoneal environment might contribute to endometriosis. Herein, we report that whereas some gut bacteria promote endometriosis, others protect against endometriosis by fermenting fiber to produce short-chain fatty acids. Specifically, we found that altered gut microbiota drives endometriotic lesion growth and feces from mice with endometriosis contained less of short-chain fatty acid and n-butyrate than feces from mice without endometriosis. Treatment with n-butyrate reduced growth of both mouse endometriotic lesions and human endometriotic lesions in a pre-clinical mouse model. Mechanistic studies revealed that n-butyrate inhibited human endometriotic cell survival and lesion growth through G-protein–coupled receptors, histone deacetylases, and a GTPase activating protein, RAP1GAP. Our findings will enable future studies aimed at developing diagnostic tests, gut bacteria metabolites and treatment strategies, dietary supplements, n-butyrate analogs, or probiotics for endometriosis.**

## Introduction

Endometriosis, in which endometrial tissue exits the uterus and implants and proliferates on peritoneal surfaces in the abdomen, afflicts ~196 million women globally, or 1 in 10 females between 12 and 52 yr of age. Half of these women experience chronic pelvic pain (1), and many experience excessive bleeding, infertility, and pain with menstruation, intercourse, bowel movements, or urination. The prevailing theory is that endometriotic lesions establish during retrograde menstruation, which 90% of women experience, expels

endometrial tissue into the peritoneal space, where it can implant on surrounding tissues such as the intestine (1). Usually, the immune system clears these cells. However, if this process fails, the endometrial cells establish lesions, which can then spread in response to inflammation and macrophage-released pro-inflammatory cytokines and growth factors (2, 3). Each of the current strategies to treat endometriosis—pain medication, hormonal therapy, surgical excision of endometriotic lesions, and hysterectomy—has negative side effects, and none can prevent recurrences (4). Thus, to develop new approaches to treat this painful disease and improve women's fertility, we need a more detailed understanding of the underlying mechanisms of and improve the women's fertility rate and health affected by this disease.

We previously reported that mice that consumed broad-spectrum antibiotics after surgical induction of endometriosis developed smaller endometriotic lesions than mice that did not consume antibiotics (5). In addition, Ata et al (6) reported that gut bacterial profiles differed between women with and without endometriosis (n = 14 per group) (6). Recently, a clinical study of human stool samples revealed that overall diversity ($\alpha$ and $\beta$) of gut microbiota was significantly higher in healthy controls than in patients with endometriosis (7). In-depth analysis suggested that 12 genera belonging to the classes Bacilli, Bacteroidia, Clostridia, Coriobacteriia, and Gammaproteobacter differed between healthy controls and endometriosis patients (7). Moreover, fecal metabolomics identified difference in gut microbiota and associated metabolites in mice with and without endometriosis (8). Although these data suggest functional crosstalk between the gut microbiota and endometriotic lesions, the mechanisms by which gut microbiota influence endometriotic lesion growth are unknown.

One mechanism by which mammalian gut bacteria affect host physiology and immunological processes (9) is by processing otherwise indigestible nutrients into biologically active metabolites (10, 11) including short-chain fatty acids (SCFAs). SCFAs such as acetate, propionate, n-butyrate, pentanoic (valeric) acid, and hexanoic (caproic) acid are used as an energy source by enterocytes or are transported into

[1]Department of Obstetrics and Gynecology, Washington University School of Medicine, St Louis, MO, USA    [2]Center for Reproductive Health Sciences, Washington University School of Medicine, St Louis, MO, USA    [3]Division of Minimally Invasive Gynecologic Surgery, Washington University School of Medicine, St Louis, MO, USA    [4]Genome Technology Access Center, McDonnell Genome Institute, Washington University School of Medicine, St Louis, MO, USA    [5]Department of Molecular and Cellular Biology, Baylor College of Medicine, Houston, TX, USA    [6]Department of Obstetrics and Gynecology, Fienberg School of Medicine, Northwestern University, Chicago, IL, USA

Correspondence: kommagani@wustl.edu

the bloodstream (12), where they can have anti-proliferative (13, 14, 15) and anti-inflammatory (16) effects on distant organs (17). For example, n-butyrate suppresses proliferation of human breast (14) and colorectal (18) cancer cells. Moreover, n-butyrate induces anti-inflammatory effects in both colonic lamina propria macrophages and bone marrow-derived macrophages (19). In addition, n-butyrate inhibits expression of the pro-inflammatory cytokines tumor necrosis factor $\alpha$ (TNF-$\alpha$) and IL-6 in lipopolysaccharide-induced macrophages (20). SCFAs primarily affect cells via two key mechanisms. First, they can activate the G-protein-coupled receptors GPR43, GPR41, and GPR109A (21), which are known to down-regulate inflammation (22, 23). Second, they can inhibit histone deacetylases (24, 25).

Here, we used a mouse injection model of endometriosis to test the hypothesis that gut microbiota–derived SCFAs influence endometriotic lesion progression. We report that mice with endometriosis have less fecal n-butyrate than those without endometriosis and that n-butyrate administration can reduce endometriotic lesion growth. Moreover, we show that n-butyrate acts through G-protein–coupled receptors (GPRs), histone deacetylases, and RAP1GAP to inhibit endometriotic lesion growth and interestingly, n-butyrate regulate RAP1GAP possibly through inhibition of HDAC1. In addition, we showed that n-butyrate reduces the level of active RAP1 in the endometriotic epithelial cells. Finally, we report that n-butyrate inhibits growth of human endometriotic cells both in vitro and in vivo in a pre-clinical mouse model.

# Results

## Gut bacteria drive lesion growth in a mouse model of endometriosis

To determine whether gut bacteria promote endometriotic lesion growth in a mouse injection model of endometriosis, we considered two possible models: germ-free or microbiota-depleted mice. We chose the microbiota-depleted model for two main reasons. First, germ-free mice have several developmental defects that microbiota-depleted mice do not have (26). This is likely because germ-free mice are sterile throughout life, whereas microbiota-depleted mice have normal microbial compositions until the time of depletion. Second, germ-free mice lack several immune functions (27) and thus are not a suitable model for inflammatory diseases such as endometriosis. In contrast, microbiota-depleted mice have nearly normal immune functions (28). We generated microbiota-depleted mice by orally gavaging mice daily with broad-spectrum antibiotics for 7 d as described previously (29, 30). Next, we induced endometriosis by dissecting the uterus out of estrogen-treated donor mice and injecting uterine fragments into the peritoneal space of control and microbiota-depleted (labeled as MD in figures) mice (31, 32) (Fig S1A). Upon examination 21 d later, MD mice had smaller and fewer endometriotic lesions that were less fluid-filled and contained fewer proliferative cells (Ki-67-positive) than control mice had (Fig S1B–G).

To determine whether the reduced endometriotic lesion growth in microbiota-depleted mice was due to altered gut bacteria, we generated microbiota-depleted mice, injected uterine fragments from control donor mice, and then orally gavaged the recipient mice with feces from mice with and without endometriosis (Fig 1A). Microbiota-depleted mice that received feces from mice with endometriosis developed endometriotic lesions that were of similar size and mass as those that developed in non-microbiota-depleted mice (Fig 1B–E, compare MD+E to control). However, microbiota-depleted mice that received feces from mice without endometriosis (MD+NE) had significantly smaller endometriotic lesions than microbiota-depleted mice that received feces from mice with endometriosis (MD+E) (Fig 1B–E). In addition, lesions in control mice and in microbiota-depleted mice that received feces from mice with endometriosis (MD+E) had thick epithelium and stroma, whereas lesions in microbiota-depleted mice that received PBS (MD+PBS) or feces from mice without endometriosis (MD+NE) had thin epithelium and stroma (Fig 1F). Finally, lesions in microbiota-depleted mice that received feces from mice with endometriosis (MD+E) had similar numbers of proliferative cells (stained with Ki-67) as lesions in control mice and more proliferative cells than lesions in microbiota-depleted mice that received feces from mice without endometriosis (MD+E) or PBS (MD+PBS) (Fig 1G). These data indicate that feces from mice with endometriosis contain a factor(s) that promotes endometriotic lesion growth or that feces from mice without endometriosis contains a factor(s) that inhibits/protects endometriotic lesion growth.

## SCFAs are reduced in feces from mice with endometriosis

Given that gut bacteria produce SCFAs that can affect host physiology, we wondered whether mice with and without endometriosis had similar concentrations of fecal SCFAs. Thus, we measured the relative concentrations of 10 SCFAs in feces from mice with (Endo) and without (Sham) endometriosis (Fig 2A). The accurate and reproducible methods for the quantification of SCFA in fecal samples using liquid chromatography-tandem mass spectrometry (LC-MS) are well established (33, 34). In sham mice, the most abundant SCFAs were acetate, propionate, and n-butyrate (Fig S2A and B). On examining all 10 SCFAs, we found that mice with endometriosis had similar concentrations of seven SCFAs (acetate, propionate, 2-methyl-butyrate, iso-valerate, 3-methly valerate, iso-caproate, and caproate) as sham (without endometriosis) mice (Fig 2A). However, mice with endometriosis had significantly lower concentrations of n-butyrate, iso-butyrate, and valerate than mice without endometriosis (Fig 2A). Apart from n-butyrate, other two most common SCFA, acetate and propionate were nonsignificantly down-regulated in the feces of mice with endometriosis (Fig 2A). These results suggest that the development of endometriosis is associated with an altered composition of gut SCFAs.

## n-butyrate inhibits endometriotic cell viability and lesion formation

We next examined the effects of the three most common SCFAs—acetate, propionate, and n-butyrate (Fig S2A and B) on endometriosis lesion growth in mice. We induced endometriosis by injecting uterine fragments from donor mice into the peritoneal

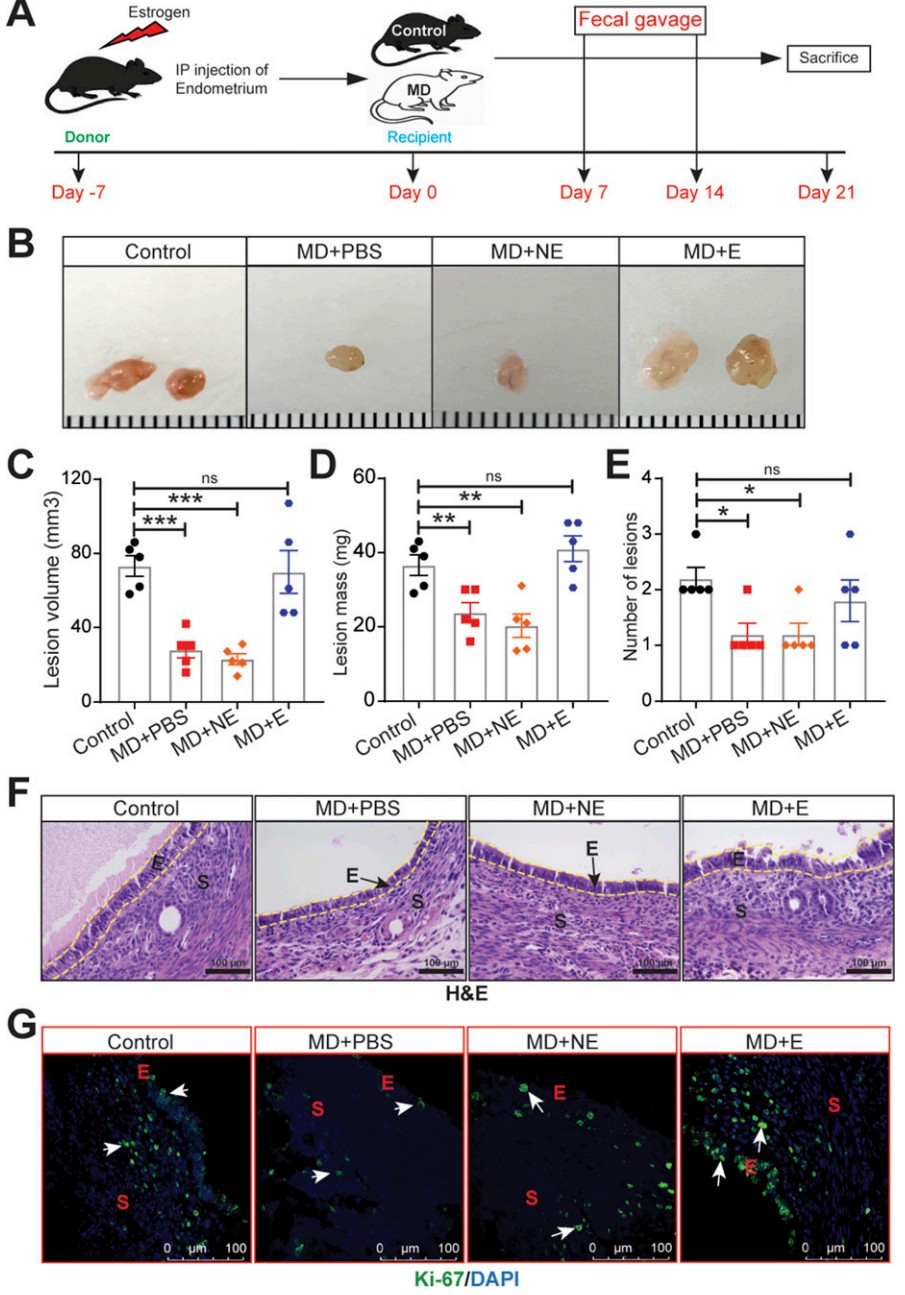

**Figure 1. Gut bacteria are required for endometriotic lesion growth in mice.**
**(A)** Schematic of experimental timeline and procedures. Mice were microbiota depleted (MD) for 7 d, then injected with uterine fragments on Day 0. Mice were orally gavaged with PBS (MD+PBS), feces from mice without endometriosis (MD+NE), or feces from mice with endometriosis (MD+E) on Days 7 and 14. **(B, C, D, E)** Representative ectopic lesion images, (C) volumes, (D) masses, and (E) number of lesions from the indicated groups, 21 d after injection of endometrial fragments. **(F, G)** Representative images of ectopic lesions from the indicated treatment groups stained with (F) hematoxylin and eosin (H & E) (yellow dashed lines demarcate the epithelium and (G) anti-Ki-67 antibody). E, epithelium; S, stroma. Data are presented as mean ± SE (n = 5 mice). Scale bar 100 μm, *$P < 0.05$; **$P < 0.01$; and ***$P < 0.001$ ns, nonsignificant.

space of recipient mice and then provided the mice with drinking water containing acetate, propionate, or n-butyrate (300 mM) (35, 36) for 21 d (Fig 2B). Mice that consumed acetate or propionate showed the modest effect on the endometriotic lesion mass and statistically nonsignificant reduction in the lesion volume (Fig 2C–F). However, mice that consumed n-butyrate developed significantly fewer and smaller lesions than mice that consumed vehicle (Fig 2C–F). In addition, lesions in mice that consumed n-butyrate had thin stroma and epithelium (Fig S3A) and fewer proliferative (Ki-67-positive) cells and macrophages (F4/80-positive cells) than lesions in mice that consumed acetate, propionate or vehicle (Fig S3B and C). These results suggest that n-butyrate, but not acetate or propionate,

inhibits endometriotic lesion growth and inflammatory cell infiltration in mice.

We next investigated the effect of n-butyrate on cells derived from human endometriotic lesions. At physiological concentrations (37), n-butyrate inhibited in vitro growth of both immortalized human endometriotic epithelial cells expressing luciferase (iHEECs/Luc) (Fig 3A) and primary human endometriotic stromal cells (HEnSCs) (Fig 3B). When we treated iHEECs/Luc (Fig S4A and B) and HEnSCs (Fig S4C and D) with acetate or propionate, it showed the modest effect on cellular proliferation at later time points only. Next, we injected iHEECs/Luc and immortalized human endometrial stromal cells expressing luciferase (iHESCs/Luc)

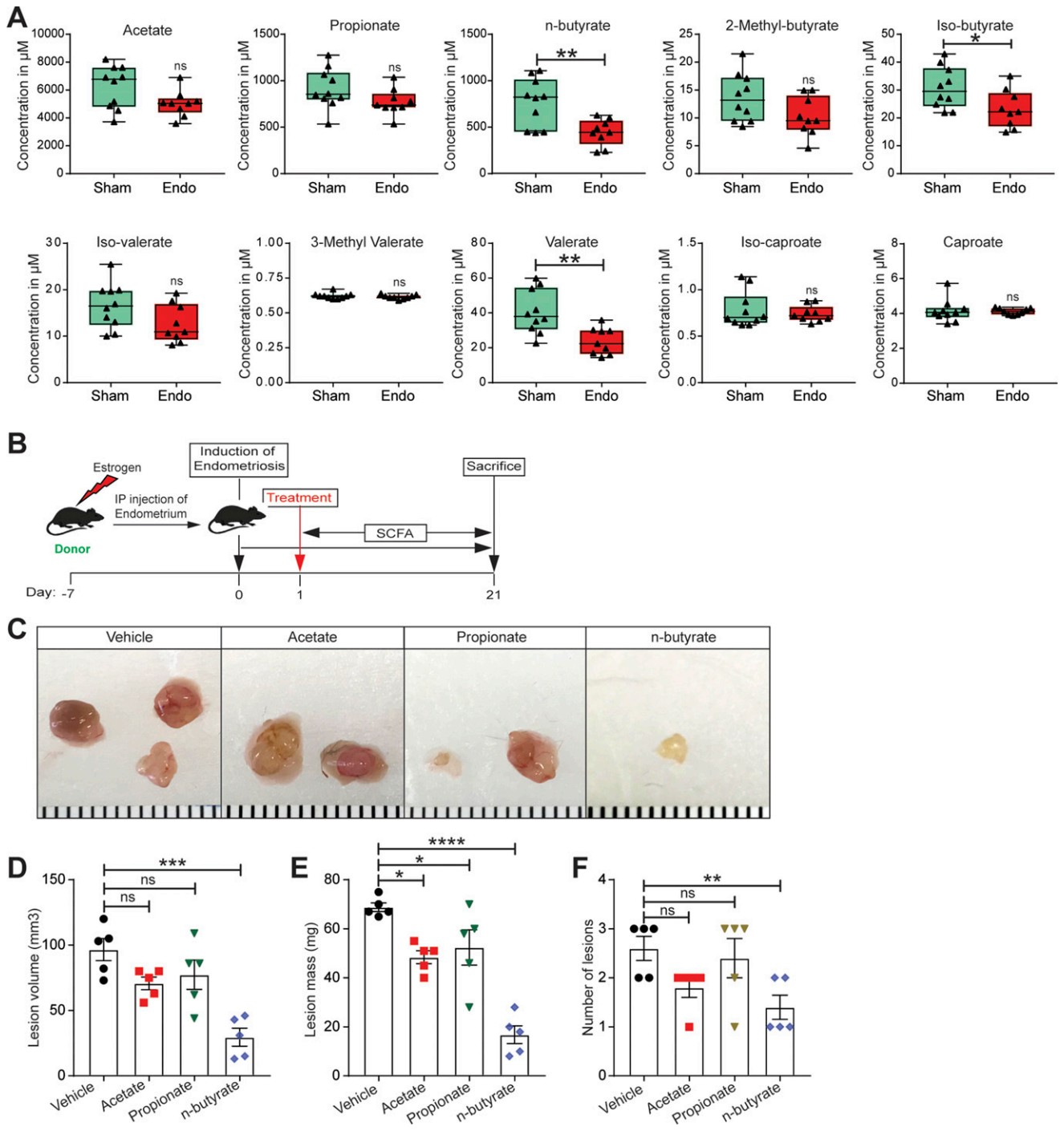

**Figure 2. n-butyrate but not acetate or propionate inhibits endometriotic lesion growth in mice.**
**(A)** The absolute concentration of indicated short-chain fatty acids in feces of mice with (Endo) and without (Sham) endometriosis. Data are presented as mean ± SE (n = 9–10 mice). **(B)** Schematic of experimental timeline and procedures. **(C, D, E, F)** Representative endometriotic lesion images, (D) volumes, (E) masses, and (F) number of lesions from the indicated groups 21 d after injection of uterine fragments. Data are presented as mean ± SE (n = 5). *$P < 0.05$, **$P < 0.01$, ***$P < 0.001$, ****$P < 0.0001$, and ns, nonsignificant.

into the peritoneal space of immunocompromised mice and provided them with drinking water containing vehicle or 300 mM n-butyrate for 21 d. Mice that consumed n-butyrate developed significantly smaller lesions than mice that consumed vehicle

(Fig 3C–F). In addition, the human cell–derived lesions in mice that consumed n-butyrate had fewer proliferative (stained for Ki-67) cells (Fig 3G and H) and macrophages (F4/80-positive cells) than lesions in mice that consumed vehicle (Fig 3I). We conclude that

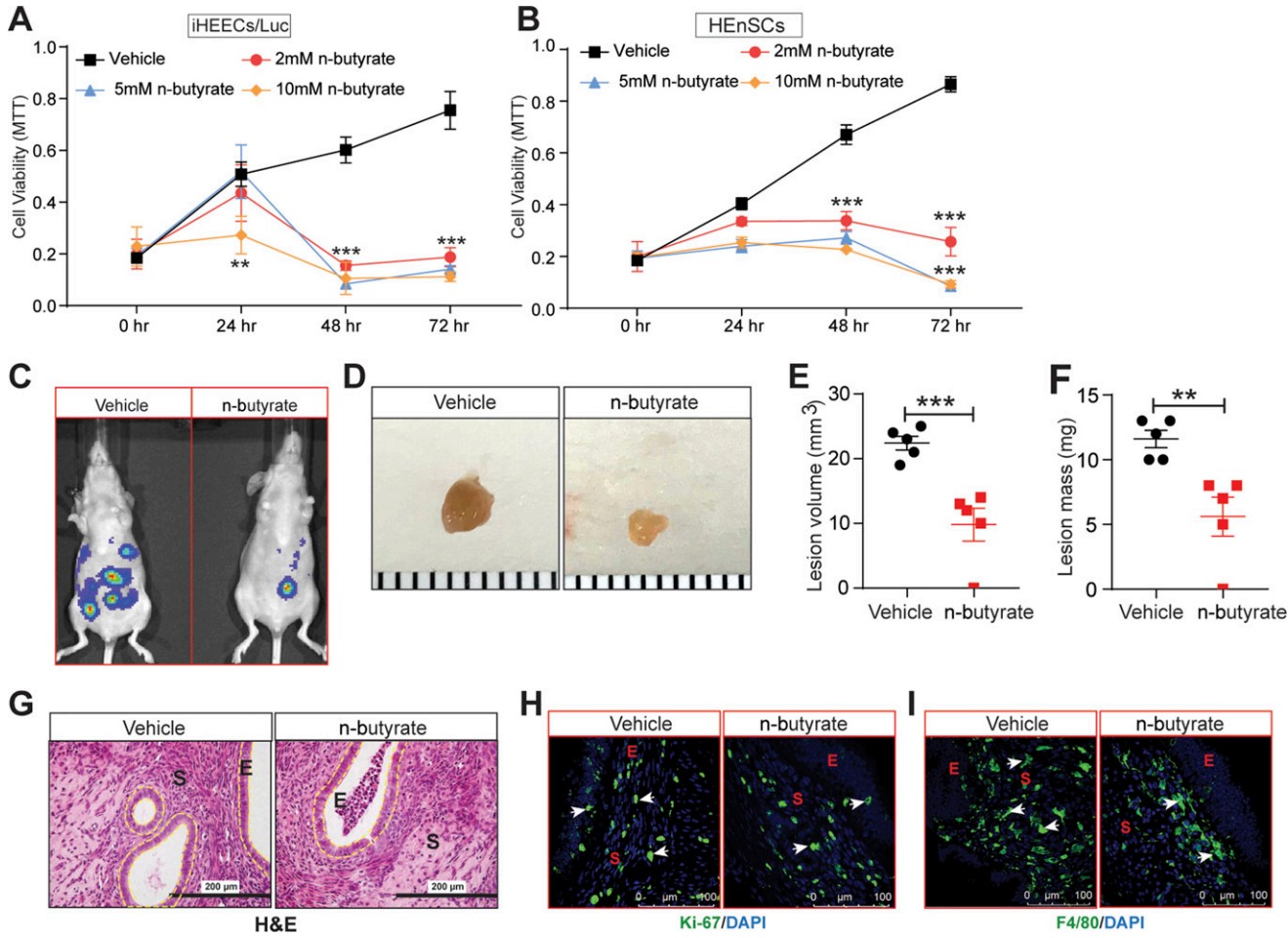

**Figure 3. n-butyrate inhibits human endometriotic lesion growth in mice.**
**(A, B)** Representative MTT cell viability assays of (A) Immortalized Human Endometriotic Epithelial Cells/Luciferase (iHEECs/Luc) and (B) primary Human Endometriotic Stromal Cells (HEnSCs) isolated from human endometriotic lesion biopsies at indicated time points and n-butyrate concentrations. Graphs represent data as mean ± SE from triplicate samples from one experiment (three experiments were conducted in total, n = 3). **(C, D)** Representative (C) bioluminescence images and (D) lesions from mice of the indicated groups 21 d after induction of endometriosis. **(E, F)** Quantitation of lesion (E) volumes and (F) masses. **(G, H, I)** Representative images of ectopic lesions from the indicated treatment groups stained with (G) hematoxylin and eosin (H & E) (yellow dashed lines demarcate the epithelium), scale bar 200 $\mu$m (H) anti-Ki-67 antibody and (I) anti-F4/80 antibody. E, epithelium; S, stroma. Data are presented as mean ± SE; (n = 5 mice per group), scale bar 100 $\mu$m; **$P < 0.01$, and ***$P < 0.001$.

n-butyrate inhibits growth of human endometriotic cells both in vitro and in vivo in a pre-clinical mouse model.

### n-butyrate inhibits endometriotic lesion growth, in part, via GPCRs

SCFAs can activate the G-protein–coupled receptors GPR43, GPR41, and GPR109A (38). As n-butyrate primarily functions through GPR43 and GPR109A (22, 23, 39, 40, 41, 42, 43, 44), we wondered whether these receptors were required for the action of n-butyrate in endometriotic cells. Thus, we pretreated iHEECs/Luc for 1 h with GPR43 antagonist GLPG0974 (100 nM) (45), GPR109A inhibitor mepenzolate bromide (MB), (100 nM) (46), or both and then treated the cells with 2 mM n-butyrate for 24 h. Whereas cells treated with n-butyrate proliferated significantly less than vehicle-treated cells, those treated with n-butyrate along with either of the GPR antagonist or inhibitors proliferated significantly more

than cells treated with n-butyrate alone. Those treated with n-butyrate plus both GPR inhibitors proliferated even more (Fig 4A). These data suggest that both GPR43 and GPR109A are required for n-butyrate–mediated inhibition of endometriotic cell growth. To confirm this finding, we transfected iHEECs/Luc with control non-targeting siRNA, siRNA targeting the gene encoding GPR43 (*FFAR2*), siRNA targeting the gene encoding GPR109A (*HCAR2*), or both targeted siRNAs. After 48 h, we treated the cells with vehicle or 2 mM n-butyrate. Knockdown of *FFAR2*, *HCAR2*, or both partially restored cell viability in n-butyrate-treated cells (Fig 4B and C). Together, these results suggest that n-butyrate signals through both GPR43 and GPR109A to prevent endometriotic cell growth.

Next, to assess the in vivo role of GPCRs in the n-butyrate–mediated suppression of endometriotic lesion growth, we induced endometriosis by injecting uterine fragments from donor mice into the peritoneal space of recipient mice. Then, we intraperitoneally

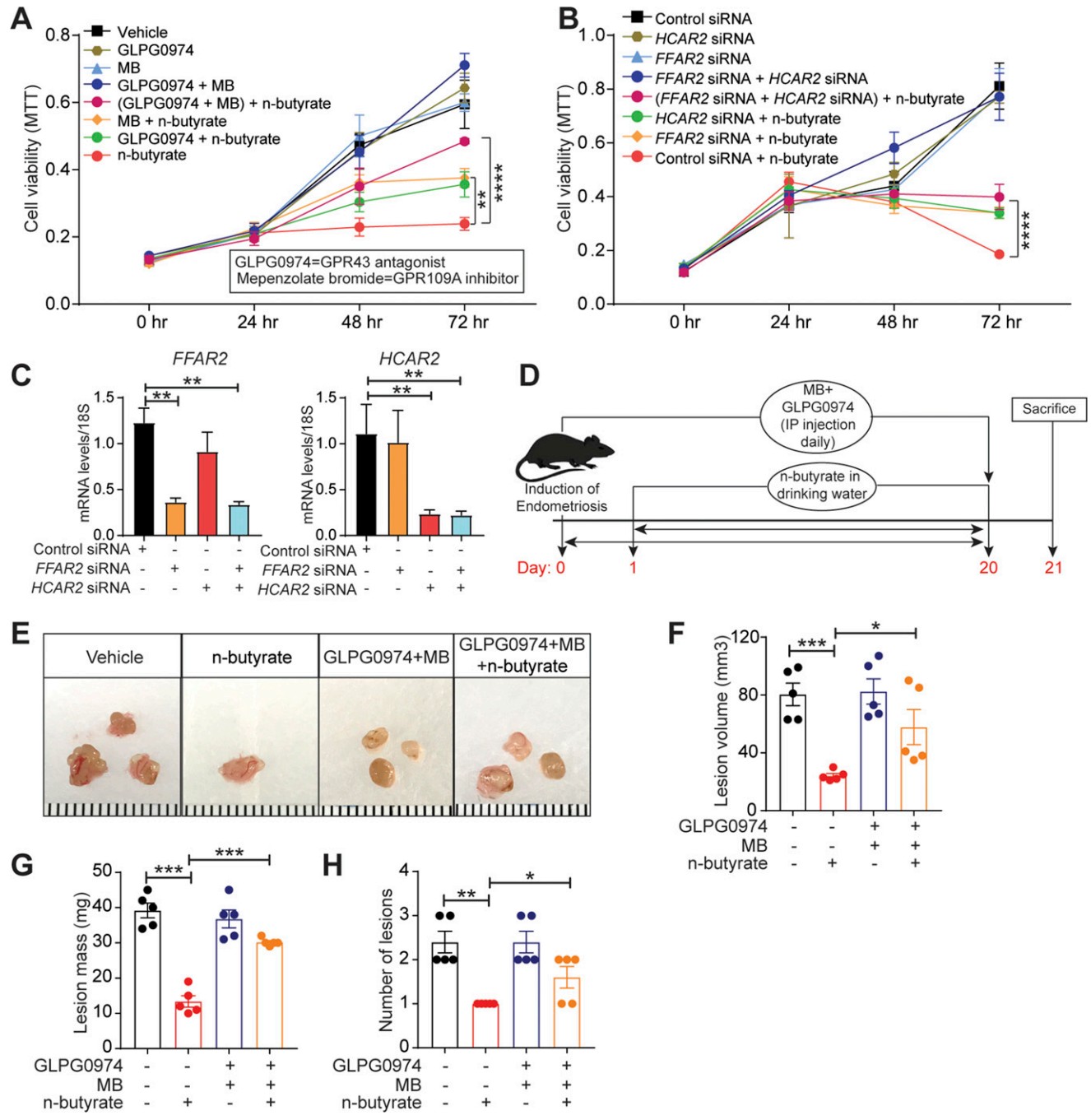

**Figure 4. n-butyrate functions through G-protein–coupled receptors (GPRs) to inhibit endometriotic lesion growth.**
**(A)** MTT cell viability assays of iHEECs/Luc treated with GPR43 antagonist GLPG0974, GPR109A inhibitor mepenzolate bromide, or both combined and treated with 2 mM n-butyrate for indicated time points. **(B)** MTT cell viability assays of iHEECs/Luc transfected with the indicated siRNAs and treated with 2 mM n-butyrate for indicated time points. **(C)** Quantitative RT-PCR of *FFAR2* and *HCAR2* in siRNA-transfected iHEECs/Luc after 48 h. The graphs in (A), (B), and (C) show representative data presented as mean ± SE from triplicate samples from one experiment (three experiments were conducted in total, n = 3). **(D)** Schematic of experimental timeline and procedures. **(E, F, G, H)** Representative images of ectopic endometriotic lesions, (F) volumes, (G) masses, and (H) numbers of lesions from the indicated groups 21 d after injection of uterine fragments. Data are presented as mean ± SE (n = 5), *P < 0.05, **P < 0.01, ***P < 0.001, ****P < 0.0001.

injected the mice with 10 mg/kg GLPG0974 (47) plus 10 mg/kg MB (48) once per day from days 0 through 21 and provided the mice with drinking water containing vehicle or 300 mM n-butyrate until day 21 (Fig 4D). Mice that received GLPG0974 plus MB plus n-butyrate developed endometriotic lesions that were larger than did the mice that received n-butyrate alone (Fig 4E–H). In addition, the mice that received the two inhibitors and n-butyrate developed endometriotic lesion that were of similar histological appearance (Fig S5A) and had

similar numbers of proliferative cells (Fig S5B) as those in control mice. These in vivo findings further support the idea that n-butyrate inhibits lesion growth by acting through both GPR43 and GPR109A receptors.

## n-butyrate inhibits HDAC, and HDAC activity is required for endometriotic cell and lesion growth

Blocking GPR43 and GPR109A only partially prevented n-butyrate–mediated inhibition of endometriotic cell growth in vitro and lesion growth in vivo, suggesting that n-butyrate also has GPCR-independent functions in endometriosis. Earlier reports suggested that n-butyrate can inhibit class I and II histone deacetylases (HDACs) (19, 24, 25, 49). HDAC activity can be assessed by measuring the amount of acetylated histone H3, and n-butyrate treatment leads to increased acetylated histone H3 in other cell types (19, 49). To determine whether n-butyrate acts as an HDAC inhibitor in iHEECs/Luc cells, we treated these cells with the pan-HDAC inhibitors trichostatin-A (TSA) and Vorinostat (SAHA); the HDAC1 and three inhibitor Entinostat (MS-275); the HDAC2 inhibitor Valproic acid (VPA); or the HDAC3 inhibitor RGFP966. Western blotting revealed that treatment with n-butyrate, TSA, SAHA, or Entinostat all increased the abundance of acetylated histone H3 (Ac-H3) to a similar extent (Fig S6A). We next measured cell viability and found that treatment with SAHA, TSA, or Entinostat inhibited viability to a greater extent than did treatment with n-butyrate (Fig S6B). Conversely, inhibition of HDAC2 and HADC3 by their respective inhibitors (VPA and RGFP966) only moderately inhibited cell viability at later time points (Fig S6B). Consistent with an idea that HDAC1 plays a role in endometriotic lesion growth, we found that it was abundantly expressed in ectopic endometriotic lesions in mice (Fig 5A).

Next, we investigated the effect of these HDAC inhibitors on growth of endometriotic lesions in mice. We induced endometriosis as described earlier and then intraperitoneally injected the mice with HDAC inhibitors once per day from days 0 through 21 (Fig 5B). Lesions from mice that received VPA (HDAC2 inhibitor) or RGFP966 (HDAC3 inhibitor) had similar volume, mass, and number as lesions from control/vehicle mice (Fig 5C–F). In contrast, lesions from mice that received n-butyrate, pan-HDAC inhibitors (TSA and SAHA), or MS-275 (HDAC1 & three inhibitor) had smaller and fewer endometriotic lesions than control/vehicle mice (Fig 5C–F). In addition, lesions in mice that consumed n-butyrate, TSA, SAHA, or MS-275 had thin stroma and epithelium (Fig 5G). We conclude that n-butyrate inhibits HDAC activity in endometriotic cells and that HDAC activity is required for endometriotic cell viability and lesion growth.

## RAP1GAP contributes to the n-butyrate–mediated inhibition of endometriotic cell viability

To further explore the mechanism by which n-butyrate inhibits endometriotic cell growth, we performed RNA-seq analysis of iHEECs/Luc treated with vehicle or n-butyrate for 24 h. As shown in Fig 6A, hierarchical clustering revealed a distinct n-butyrate–dependent transcriptome in iHEECs/Luc cells. Using a 2.0-fold cutoff and Benjamini–Hochberg false discovery rate of <0.05 threshold for inclusion, we identified 1,830 genes that were differentially expressed between vehicle- and n-butyrate–treated iHEECs/Luc (Fig S7A

and B and Table S1). Gene Ontology (GO) enrichment analysis revealed that n-butyrate up-regulated expression of genes involved in several biological processes including synaptic signaling, cation transmembrane transport, ion and chemical homeostasis, and cell–cell signaling, and down-regulated expression of genes involved in chromosome organization, chromatin organization, histone modification, covalent chromatin modification, and cellular response to DNA damage. In addition, the top 25 pathways containing the most significantly up-regulated genes in n-butyrate–treated cells included calcium signaling, metabolic pathways, RAP1 signaling, etc. (Fig S7C).

We chose to focus on the RAP1 signaling pathway for several reasons. First, this pathway plays a key role in cell proliferation, growth, adhesion, and motility, and RAP1 is a central regulator of tumor cell migration and invasion (50). RAP1 is active when bound to GTP, and its activity is regulated by GTP exchange factors, which activate it, and GTPase activating protein (RAP1GAP), which inactivates it. Consistent with its ability to inactivate RAP1, RAP1GAP is a tumor suppressor in several human cancers, including endometrial (51), thyroid (52, 53), pancreas (54), colon (55), melanomas (56), prostate (57), and head and neck carcinomas (58). Second, we found that expression of several genes in the RAP1 pathway were differentially expressed between vehicle and n-butyrate–treated cells (Fig S8). Third, RAP1GAP was one of the top genes up-regulated by n-butyrate in our RNA-Seq analysis (Table S1). Fourth, from publicly available Gene Expression Omnibus (GEO) datasets (GSE6364), we found that the RAP1GAP raw expression score was significantly lower in endometriotic lesions than in endometrial tissue from healthy control women (59) (Fig 6B). Furthermore, we confirmed by qRT-PCR that expression of RAP1GAP, but not RAP1GAP2 or RAP1GDS was up-regulated in iHEECs/Luc cells exposed to n-butyrate (Fig 6C). In contrast to n-butyrate, acetate and propionate had no effect on RAP1GAP (Fig S9A) RAP1GAP2 (Fig S9B) and RAP1GDS (Fig S9C) expression in iHEECs/Luc cells. Subsequently, we analyzed the level of active RAP1 in the iHEECs/Luc treated with 2 mM n-butyrate for 24 h. Active RAP1 detection kit based analysis revealed that n-butyrate significantly reduced the level of active RAP1 (Fig 6D, left panel). In contrast, the total level of RAP1 remains unchanged (Fig 6D, right panel). Finally, we confirmed that the level of RAP1GAP was equally induced by both n-butyrate and Entinostat (MS-275), suggesting that n-butyrate might induce the RAP1GAP through inhibition of HDAC1 (Fig S10A). Furthermore, treatment of n-butyrate or MS-275 had no effect on the induction of RAP1GAP2 (Fig S10B) and RAP1GDS (Fig S10C).

Given the role of RAP1GAP as a tumor suppressor and our data indicating that its expression is up-regulated by n-butyrate, we wondered whether RAP1GAP is required for n-butyrate–mediated suppression of cellular viability. To test this idea, we knocked down RAP1GAP expression in iHEECs/Luc cells for 48 h and then treated them with vehicle or 2 mM n-butyrate. Knockdown of RAP1GAP in untreated cells had no effect on cell viability. However, n-butyrate–treated cells in which RAP1GAP was knocked down proliferated significantly more than n-butyrate–treated cells transfected with control siRNA (Fig 6E). We conclude that n-butyrate inhibits endometriotic cell growth, in part, by inducing expression of RAP1GAP, resulting in inactivation of the pro-growth RAP1 signaling pathway.

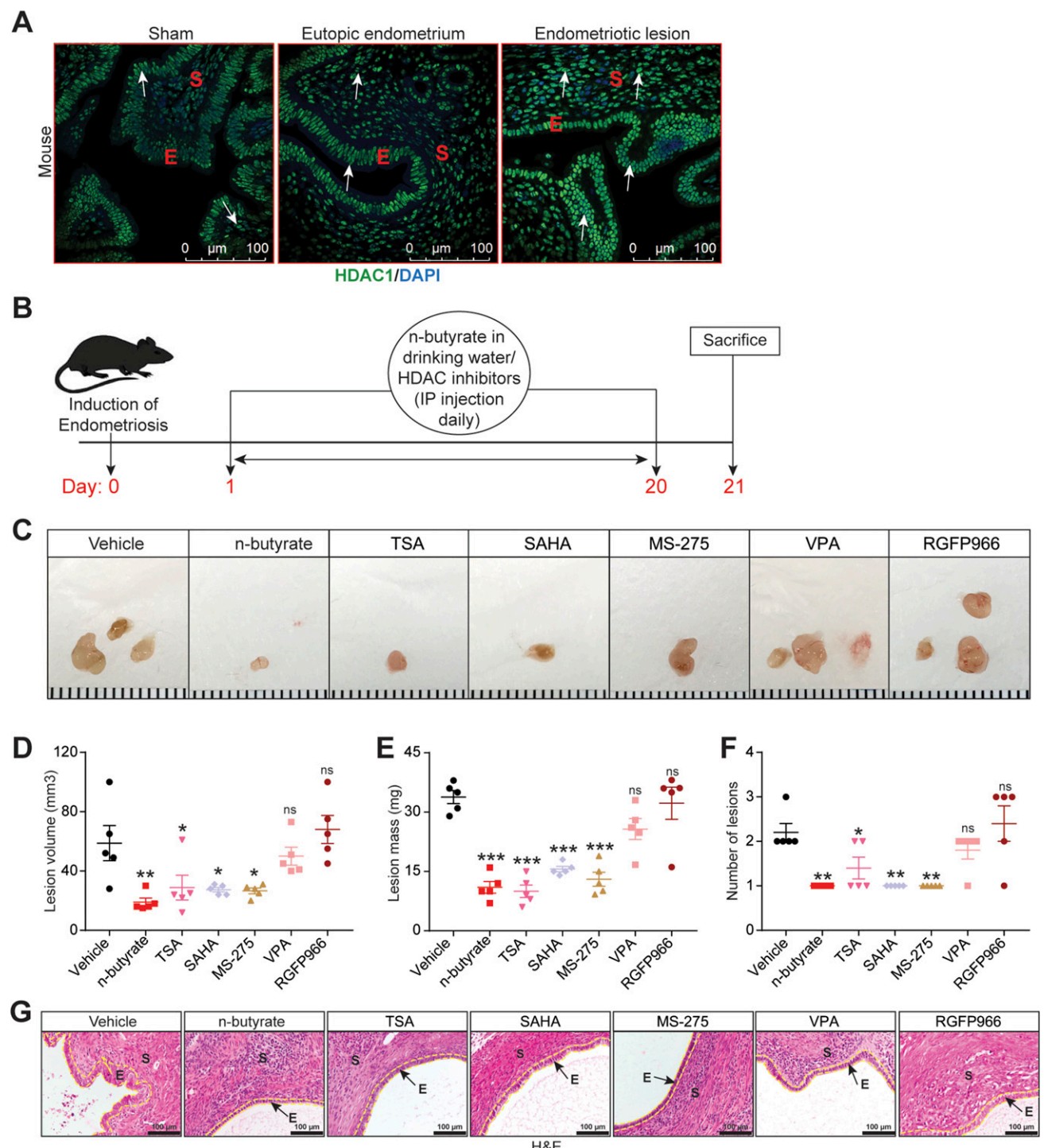

**Figure 5.  n-butyrate inhibits HDAC activity, which is required for endometriotic cell growth.**
**(A)** Representative images of eutopic endometrium and ectopic lesions of mice stained with anti-HDAC1 antibody from the indicated groups. E, epithelium; S, stroma. White arrow indicates the HADC1-positive cells. Scale bar 100 μm. **(B)** Schematic of experimental timeline and procedures. **(C, D, E, F)** Representative images of endometriotic lesions, (D) volumes, (E) masses, and (F) number of lesions from the indicated treatment groups 21 d after injection of uterine fragments. **(G)** Representative images of ectopic lesions stained with hematoxylin and eosin (H & E) from the indicated treatment groups (yellow dashed lines demarcate the epithelium). E, epithelium; S, stroma. Scale bar 100 μm. Data are presented as mean ± SE (n = 5), *P < 0.05, **P < 0.01, ***P < 0.001, and ns, nonsignificant.

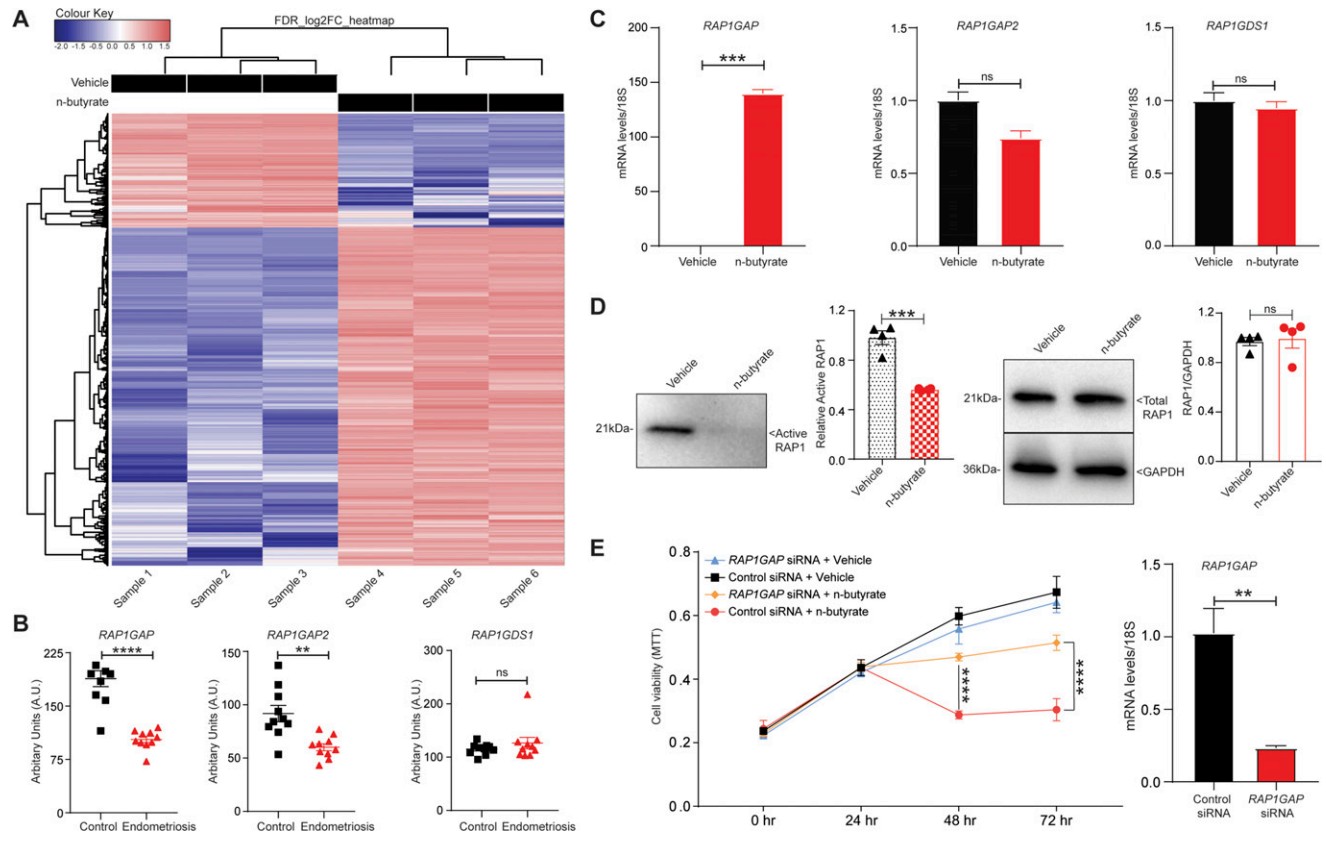

**Figure 6. RAP1GAP mediates n-butyrate–driven endometriotic cell growth inhibition.**
**(A)** Heat map of transcripts differentially expressed between vehicle- and n-butyrate–treated iHEECs/Luc with cutoff of FDR < 0.05 and logFC >2.0; n = 3 each group. **(B)** Relative raw abundance of *RAP1GAP*, *RAP1GAP2*, and *RAP1GDS* transcripts from microarray analysis within a publicly available GEO dataset (GSE6364). Data are presented as mean ± SE (n = 10). **(C)** Relative abundance of *RAP1GAP*, *RAP1GAP2*, and *RAP1GDS* transcripts in iHEECs/Luc treated with 2 mM n-butyrate for 24 h; n = 3 each group. **(D)** Relative abundance of active RAP1 (GTP-bound) and total RAP1, in iHEECs/Luc treated with 2 mM n-butyrate for 24 h; n = 4 each group. **(E)** MTT assay of iHEECs/Luc transfected with control or *RAP1GAP* siRNA and then treated with 2 mM n-butyrate for indicated times. Data are presented as the mean ± SE from triplicate samples from one experiment (three experiments were conducted in total). The graph on the right depicts quantitative RT-PCR–based confirmation of *RAP1GAP* knockdown in iHEECs/Luc after 48 h of siRNA transfection. \*\*P < 0.01, \*\*\*P < 0.001, \*\*\*\*P < 0.0001, and ns, nonsignificant.

# Discussion

Together, our data support the following model. First, endometriosis alters the gut microbiome, resulting in reduced production of the SCFA, n-butyrate. Second, n-butyrate, but not acetate or propionate, inhibits endometriotic lesion growth. Third, n-butyrate reduces endometriotic growth by at least three potentially overlapping mechanisms: activating GPR43 and GPR109A, inhibiting histone deacetylase activity, and activating expression of RAP1GAP, which inactivates the pro-growth RAP1 signaling pathway.

Several recent studies revealed a correlation between microbiota and endometriosis pathogenesis (6, 31, 60). For example, women with endometriosis were more likely than women without endometriosis to have uterine microbial dysbiosis (6, 61, 62, 63). Moreover, previous work from our laboratory and others showed that mice and women with endometriosis had altered gut microbial communities (5, 6, 7, 31, 64). This idea is consistent with our fecal microbiota transplant experiments, in which only fecal samples from mice with endometriosis restored lesion growth in microbiota-depleted mice. Moreover, our findings support the idea that gut bacteria, as opposed to bacteria elsewhere in the body, play a role in endometriotic lesion development. Given that women with endometriosis have increased susceptibility to inflammatory bowel disease (65), this altered gut bacteria in fact may link endometriosis progression and colonic disease.

SCFAs are the by-products of bacterial fermentation of dietary fiber and are generally beneficial (66, 67). SCFAs act as signaling molecules and regulate several host biological processes, including metabolism, immune function, and cellular proliferation (68, 69). The most abundant SCFAs in humans are acetate, propionate, and n-butyrate (70, 71, 72, 73). Although most SCFAs are used by colon epithelial cells, SCFAs do enter the blood and reach peripheral tissues (74). Generally, SCFAs are detected in serum and urine (75), although at much lower concentrations than in the gut and feces (76). Nonetheless, SCFAs act at extra-intestinal sites and can alleviate diabetes (77, 78) asthma (79), bone loss (80, 81), and obesity (82). Treatment of HDAC inhibitor romidepsin inhibits the human endometriotic cell proliferation, and VEGF expression (83, 84). Our findings that n-butyrate inhibited HDAC activity and that HDAC activity is required for endometriotic cell growth are consistent with

studies reporting elevated HDAC1 expression in lesions from women with endometriosis (85, 86). Thus, future efforts could be directed at delivering HDAC inhibitors, bacteria engineered to over-produce n-butyrate (87, 88), n-butyrate producing *Lactobacillus* strains (81), or n-butyrate analogs to treat endometriosis.

In many cell types, SCFAs function by activating G-protein coupled receptors (19, 89, 90, 91). Consistent with the fact that n-butyrate primarily activates GPR43 (40, 41, 42) and GPR109A (22, 23) in other tissues; we found that both GPR43 and GPR109A were required for n-butyrate–mediated inhibition of endometriotic cell and lesion growth. Given the anti-inflammatory role of these, G-protein coupled receptors (91, 92), we plan to use GPR43 and GPR109A null mice or double knockouts to determine whether n-butyrate acts through these receptors to inhibit the inflammation associated with endometriosis. Our findings that n-butyrate inhibited endometriotic cell growth, in part, via RAP1GAP are consistent with the role of RAP1GAP as a tumor suppressor in multiple cancers. Given our data, we are especially interested to determine whether GPR43, GPR109A, and HDACs are involved in n-butyrate–mediated regulation of RAP1GAP expression. Such work may have implications beyond endometriosis and could deepen our understanding of the mechanisms by which n-butyrate affects growth of many tumor types.

In summary, our findings demonstrate that the bacteria-derived metabolite n-butyrate reduces endometriotic lesion growth. As production of SCFAs is dependent on both the type of gut bacteria and dietary fiber intake (93), new avenues to prevent endometriosis could include diet regimens, n-butyrate analogs, probiotics with n-butyrate-producing bacteria, or n-butyrate-containing dietary supplements. Finally, future work should be directed at determining whether women with endometriosis have lower fecal n-butyrate concentration than do healthy controls. If so, such a finding could lead to development of a simple diagnostic or predictive tool for endometriosis.

# Materials and Methods

### Animal studies

Mouse studies were performed according to a protocol (number 2019-1079) approved by the Washington University School of Medicine Institutional Animal Care and Use Committee. Mice (C57BL/6, Taconic; and immunocompromised nude, NU-FOXN1[NU], Cat. no. 088-CRL, Charles River Lab) were maintained in standard 12-h light/dark conditions and provided ad libitum access to food and water.

### Microbiota depletion with antibiotics

Mice (9–10 wk of age) were orally gavaged every 12 h for 7 d with a cocktail containing 100 mg/kg ampicillin, 50 mg/kg vancomycin, 100 mg/kg neomycin, 100 mg/kg metronidazole, and 1 mg/kg amphotericin-B. A gavage volume of 10 ml/kg body weight was delivered with a stainless steel tube without sedation (28, 29). A fresh antibiotic mixture was prepared every 3 d. The control mice were gavaged with a similar volume of water.

### Heterologous injection endometriosis model

Donor mice were subcutaneously injected with estradiol benzoate (3 μg/mouse or 100 μg/kg) on day -7 (31, 32). On day 0, donor mice were euthanized (one donor mouse for every two recipients), and uteri were removed, placed in a Petri dish containing warm saline, and cut longitudinally with scissors (94, 95). Endometrial tissue from each uterine horn was mechanically disrupted to produce two suspensions in which the maximal diameter of any piece of endometrial tissue was less than 1 mm. Each suspension (0.4 ml) was intraperitoneally injected (96) into a recipient mouse with a 1-ml syringe and a 25-g needle (97). For the sham condition, a similar procedure was performed, except mice were injected with saline. After 21 d, mice were euthanized by cervical dislocation. The abdominal cavity was immediately opened, and lesions were excised, counted, measured, weighed, and processed for histology and immunofluorescence (31, 32). 1.5-ml tubes were used to collect a fresh fecal sample from the mice, by holding mice on the cage and little pressure was applied on the back of the mice, and they defecate into a tube, two to three times were tried to collect feces from individual mice.

### Oral gavage with feces

Fecal pellets from mice were frozen at –80°C immediately after collection as reported previously (5, 98). On the day of transplantation, fecal pellets were resuspended in PBS (1 fecal pellet/0.1 ml of PBS), and 200 μl of pooled fecal material was given by oral gavage on days 7 and 14 after induction of endometriosis.

### Measuring SCFAs by mass spectrometry

The concentrations of SCFAs were measured in the metabolomics core at Baylor College of Medicine. Procedures for sample preparation, extraction, and analysis by derivatization were performed as previously described (99). Briefly, 500 μl acetonitrile was added to each fecal sample or to liver tissue (quality controls), samples were homogenized, and supernatant was collected. To 40 μl of supernatant, 20 μl of 200 mM 12C6-3NPH and 120 mM 1-Ethyl-3-(3-dimethylaminopropyl) carbodiimide (EDC) were added, and the mixture was incubated for 30 min at 40°C. The mixture was then cooled and made up to 1.91 ml with 10% aqueous acetonitrile, and 10 μl of this solution was injected into a liquid chromatography tandem mass spectrometer (Agilent Technologies) (100). Agilent Mass Hunter workstation software was used to analyze chromatograms, and the peak area was integrated depending on the retention time. The concentration of each measured metabolite was calculated from normalized data (101, 102, 103, 104). Two sample *t* tests were conducted to assess differences in concentrations of each metabolite (101, 102, 103, 104).

### Treatment of mice with SCFAs, GPR modulators, and HDAC inhibitors

From day 0 to day 21 after endometriosis induction, mice were provided drinking water containing 300 mM sodium acetate (36), sodium propionate (36), or sodium butyrate (19) (all from Sigma-Aldrich) (36). These solutions were changed every week. The GPR43

antagonist GLPG0974 (10 mg/kg) and GPR109A inhibitor MB (10 mg/kg) were dissolved in dimethyl sulfoxide (vehicle) with 30% PEG-300 (Sigma-Aldrich) and administered via daily intra-peritoneal injection. Trichostatin-A (1 mg/kg), SAHA (25 mg/kg), MS-275 (20 mg/kg), valproic acid (500 mg/kg), and RGFP0966 (25 mg/kg) (all from Sigma-Aldrich) were dissolved in dimethyl sulfoxide with 30% PEG-300 and administered from day 0 to day 21 via daily intra-peritoneal injection. Similar amount of dimethyl sulfoxide with 30% PEG-300 was administered as vehicle.

## Isolation of stromal cells from human endometriotic lesions

Human endometriotic tissues were obtained from women under a protocol (IRB ID #: 201807160) approved by the Washington University Institutional Review Board. All participants were recruited through the Washington University online classified section and local newspaper ads. Eligible participants signed an Informed Consent and Authorization form. Participants were excluded if they had used probiotics, antibiotics, or any anti-inflammatory drugs within 2 wk before surgery or had a history of uterine fibroids, polycystic ovarian syndrome, or endometrial cancer. Ectopic endometriotic lesions and eutopic endometrial biopsies were collected from women undergoing endometriosis surgery. Human endometriotic stromal cells were isolated from biopsies as previously described (105). In brief, cells were cultured in DMEM/F12 (Thermo Fisher Scientific) containing 10% fetal bovine serum and 1% antibiotic and antimycotics in a humidified atmosphere with 5% $CO_2$ and 95% air at 37°C. All experiments were carried out with human endometriotic stromal cells isolated from at least three participants. Dr. Serdar Bulun from Northwestern University, Feinberg School of Medicine, generously provided one additional human endometriotic stromal cell lines.

## siRNA transfection

Immortalized Human Endometrial Stromal Cells/Luciferase (iHESCs/Luc) and Immortalized Human Endometriotic Epithelial Cells/Luciferase (iHEECs/Luc) (both cell lines generously provided by Dr. Sang Jun Han from Baylor College of Medicine (106)) were separately maintained in DMEM/F12 containing 10% FBS, 100 U/ml penicillin, 100 mg/ml streptomycin, and 2.5 mg/ml amphotericin-B in humidified condition with 5% $CO_2$ and 95% air at 37°C. The iHEECs/Luc cells are derived from an ovarian endometrioma lesion as described previously (107). The medium was changed every other day. The iHEECs/Luc were plated in six-well culture plates and treated in triplicate with Lipofectamine 2000 transfection reagent and 60 pmol of non-targeting siRNA (D-001810-10-05) or siRNAs targeting *FFAR2* (L-005574-00-0005), *HCAR2* (L-006688-02-0005), or *RAP1GAP* (L-019706-00-0005) (GE Healthcare Dharmacon Inc.), as described previously (105, 108). After 48 h, cells were treated with 2 mM sodium butyrate in complete growth media.

## Xenotransplantation of human endometrial cells

This model of endometriosis was generated in athymic nude mice (Charles River) as described previously (106). Briefly, 2 d before the day of transplantation, mice were ovariectomized, and a sterile 60-d release pellet containing 0.36 mg of 17-$\beta$ estradiol (Innovative Research of America) was implanted. On the day of transplantation, iHESCs/Luc and iHEECs/Luc cells were trypsinized with 0.05% trypsin–EDTA, and 2 × $10^6$ iHESCs/Luc and 2 × $10^6$ iHEECs/Luc cells were combined in 10 ml of DMEM/F12, pelleted, washed, resuspended in 100 $\mu$l of DMEM/F12, and mixed with 100 $\mu$l of Matrigel (BD Biosciences). The cell suspension/Matrigel mixture (200 $\mu$l) was intraperitoneally injected into the mice on the midventral line just caudal to the umbilicus. After 21 d, mice were injected with D-Luciferin, and bioluminescence images of each mouse were collected with an in vivo image analysis system. Mice were then euthanized and endometriotic lesions were collected. Endometriotic lesion volumes (cubic millimetre) were measured with a Vernier Caliper.

## Hematoxylin and Eosin staining

Tissues were fixed in 4% paraformaldehyde, embedded in paraffin, and then sectioned (5 $\mu$m) with a microtome (Leica Biosystem). Tissue sections were deparaffinized, rehydrated, and stained with Hematoxylin and Eosin as described previously (5). All the histology was performed on three sections from each lesion of individual mice, and one representative section image is shown in the respective figures.

## Immunofluorescence

Formalin-fixed and paraffin-embedded sections were deparaffinized in xylene and rehydrated in an ethanol gradient, and antigen was retrieved after boiling in citrate-buffer (Vector Laboratories Inc.). After blocking with 2.5% goat serum (Vector laboratories) diluted in PBS for 1 h at room temperature, sections were incubated overnight at 4°C with primary antibodies (Table S2) diluted in 2.5% normal goat serum. After washing with PBS, sections were incubated with Alexa Fluor 488–conjugated secondary antibodies (Life Technologies) for 1 h at room temperature, washed, and mounted with ProLong Gold Antifade Mountant with DAPI (Thermo Fisher Scientific).

## Cell viability assays

Cell viability was determined by performing the 3-(4,5-Dimethylthiazol-2-yl)-2,5-Diphenyltetrazolium Bromide (MTT) assay (Promega) according to the manufacturer's instructions. Briefly, iHEECs/Luc or primary human endometriotic stromal cells were counted and plated in 96-well plates, and relative viability rates were evaluated at the indicated time points after treatment with n-butyrate, 100 nM GLPG0974, 100 nM MB, 20 $\mu$M SAHA, 500 nM TSA, 3 $\mu$M MS-275, 3 mM Valproic acid, or 10 $\mu$M RGFP0966 (all from Sigma-Aldrich). For the GPRs study, the cells were pre-treated for 1 h with 100 nM GLPG0974, 100 nM MB, or both, then treated with 2 mM n-butyrate for 24 h. For knockdown experiments, after 48 h of siRNA transfection, iHEECs/Luc were re-plated in 96-well plates at 5 × $10^3$ cells per well. After 24 h, cells were treated with 2 mM sodium butyrate (Sigma-Aldrich) for 0, 24, 48, or 72 h. At each time point, cell viability was determined by the MTT assay. In all cases, 15 $\mu$l of MTS (dye solution) reagent (Promega) was added to each well and

incubated for another 2 h. After addition of 100 $\mu$l of Solubilization Solution, absorbance was measured at 570 nm with 650 nm as a reference wavelength in a 96-well plate reader. The experiments were performed three times each with three to five technical replicates.

## Transcription analysis

Cells were lysed in lysis buffer, and total RNA was isolated with the Purelink RNA mini kit (Invitrogen) according to the manufacturer's instructions. RNA was quantified with a Nano-Drop 2000 (Thermo Fisher Scientific). Then, 1 $\mu$g of RNA was reverse transcribed with the High-Capacity cDNA Reverse Transcription Kit (Thermo Fisher Scientific). The amplified cDNA was diluted to 10 ng/$\mu$l, and qRT-PCR was performed with primers listed in Table S3 and Fast Taqman 2X mastermix (Applied Biosystems/Life Technologies) on a 7500 Fast Real-Time PCR system (Applied Biosystems/Life Technologies). The delta–delta cycle threshold method was used to normalize expression to the reference gene 18S.

## Detection of active RAP1

Active RAP1 (GTP-bound) was detected using an active RAP1 detection kit, Cat, no. #8818; Cell Signaling Technology Inc. All steps were performed according to the manufacturer's instructions. Briefly, 750 $\mu$g of protein lysate from iHEECs/Luc cells treated with vehicle or 2 mM n-butyrate for 24 h were mixed to the GST-RalGDS-RBD in a spin cup inserted in the collection tube. Subsequently, the spin cups were incubated at 4°C for 1 h with gentle rocking. The GTP-bound RAP1 (active RAP1) protein was eluted by adding the reducing sample buffer. Finally, the eluted samples proceeded for Western blotting.

## Western blotting

Protein lysates (40 $\mu$g per lane) were loaded on a 4–15% SDS–PAGE gel (Bio-Rad), separated in 1X Tris-Glycine Buffer (Bio-Rad), and transferred to Polyvinylidene fluoride (PVDF) membranes (Millipore) via a wet electro-blotting system (Bio-Rad), all according to the manufacturer's directions and as described previously ([109]). PVDF membranes were blocked for 1 h in 5% non-fat milk (Bio-Rad) in Tris-buffered saline containing 0.1% Tween-20 (TBS-T; Bio-Rad), then incubated overnight at 4°C with antibodies listed in Table S2 in 5% BSA in TBS-T. Blots were then probed with anti-Rabbit IgG conjugated with horseradish peroxidase (1:5,000; Cell Signaling Technology) in 5% BSA in TBS-T for 1 h at room temperature. Signal was detected with the Immobilon Western Chemiluminescent HRP Substrate (Millipore), and blot images were collected with a Bio-Rad ChemiDoc imaging system.

## RNA sequencing and analysis

The iHEECs/Luc were treated with 2 mM n-butyrate for 24 h and RNA was isolated as mentioned above. The experiment was repeated three times with minimum three technical replicates. Total RNA integrity was determined using Agilent Bioanalyzer or 4200 Tapestation. Library preparation was performed with 500 ng–1 $\mu$g of total RNA. Ribosomal RNA was removed by an RNase-H method using RiboErase kits (Kapa Biosystems). mRNA was then fragmented

in reverse transcriptase buffer and heating to 94°C for 8 min. mRNA was reverse transcribed to yield cDNA using SuperScript III RT enzyme (Life Technologies, per manufacturer's instructions) and random hexamers. A second strand reaction was performed to yield ds-cDNA. cDNA was blunt ended, had an "A" base added to the 3' ends, and then had Illumina sequencing adapters ligated to the ends. Ligated fragments were then amplified for 12–15 cycles using primers incorporating unique dual index tags. Fragments were sequenced on an Illumina NovaSeq-6000 using paired end reads extending 150 bases. Basecalls and demultiplexing were performed with Illumina's bcl2fastq2 software. RNA-seq reads were then aligned to the Ensembl release 76 primary assembly with STAR version 2.5.1a. Gene counts were derived from the number of uniquely aligned unambiguous reads by Subread: featureCount version 1.4.6-p5. Isoform expression of known Ensembl transcripts was estimated with Salmon version 0.8.2. Sequencing performance was assessed for the total number of aligned reads, total number of uniquely aligned reads, and features detected. The ribosomal fraction, known junction saturation, and read distribution over known gene models were quantified with RSeQC version 2.6.2.

All gene counts were then imported into the R/Bioconductor package EdgeR, and TMM normalization size factors were calculated to adjust for differences in library size. The TMM size factors and the matrix of counts were then imported into the R/Bioconductor package Limma. Weighted likelihoods based on the observed mean–variance relationship of every gene and sample were then calculated for all samples with the voomWithQualityWeights. The performance of all genes was assessed with plots of the residual SD of every gene to their average log-count with a robustly fitted trend line of the residuals. Differential expression analysis was then performed to identify differences between conditions, and the results were filtered for only those genes with Benjamini–Hochberg false-discovery rate adjusted $P$-values less than or equal to 0.05.

Global perturbations in known GO terms, MSigDb, and Kyoto Encyclopedia of Genes and Genomes (KEGG) pathways were detected with the R/Bioconductor package GAGE to test for changes in $\log_2$-fold-change expression between the genes within a gene set over those in the background. The R/Bioconductor package heatmap3 was used to display heat maps across groups of samples for each GO or MSigDb term with a Benjamini–Hochberg false-discovery rate adjusted $P$-value less than or equal to 0.05. Perturbed KEGG pathways with $P$-values less than or equal to 0.05 were rendered as annotated KEGG graphs with the R/Bioconductor package Pathview.

To identify differentially expressed genes, the raw counts were variance stabilized with the R/Bioconductor package DESeq2 and then analyzed via weighted gene correlation network analysis with the R/Bioconductor package WGCNA. Briefly, all genes were correlated across each other by Pearson correlations and clustered by expression similarity into unsigned modules using a power threshold empirically determined from the data. An eigengene was then created for each de novo cluster, and its expression profile was correlated across all coefficients of the model matrix. Because these clusters of genes were created by expression profile rather than known functional similarity, the clustered modules were given the names of random colors such that grey was the only predefined module and contained genes that did not cluster well with others. These de novo clustered genes were then tested for functional enrichment of known GO terms with hypergeometric tests available

in the R/Bioconductor package clusterProfiler. Significant terms with Benjamini–Hochberg adjusted *P*-values less than 0.05 were then collapsed by similarity into clusterProfiler category network plots to display the most significant terms for each module of hub genes to interpolate the function of each significant module. The information for all clustered genes for each module were then combined with their respective statistical significance results from Limma to determine whether or not those features were also found to be significantly differentially expressed.

### Statistical analysis

A two-tailed paired *t* test was used to analyze data from experiments with two experimental groups and ANOVA by nonparametric alternatives was used for multiple comparisons to analyze data from experiments containing more than two groups. $P < 0.05$ was considered significant. All data are presented as mean ± SE. GraphPad Prism 8 software was used for all statistical analyses.

# Data Availability

Transcriptome data from this study are available at GEO under accession number GSE184431.

# Supplementary Information

# Acknowledgements

We thank Dr. Deborah J. Frank (Department of Obstetrics and Gynecology, Washington University) for assistance with manuscript editing and Alma Jackson (Department of Obstetrics and Gynecology, Washington University) for technical expertise. We also thank the Genome Technology Access Center in the Department of Genetics at Washington University School of Medicine for service with genomic analysis. The Center is partially supported by National Cancer Institute (NCI) Cancer Center Support Grant #P30 CA91842 to the Siteman Cancer Center and by Institute for Clinical and Translational Science (ICTS)/Clinical and Translational Sciences Award (CTSA) Grant# UL1TR002345 from the National Center for Research Resources, a component of the National Institutes of Health (NIH), and NIH Roadmap for Medical Research. This work was funded, in part, by National Institutes of Health/ National Institute of Child Health and Human Development grants R01HD102680, R01HD065435, and R00HD080742 to R Kommagani. The metabolomics core at Baylor College of Medicine is supported by the CPRIT Core Facility Support Award RP170005 and NCI Cancer Center Support Grant P30CA125123.

### Author Contributions

SB Chadchan: conceptualization, data curation, formal analysis, validation, investigation, visualization, methodology, and writing—original draft, review, and editing.
P Popli: formal analysis and writing—original draft, review, and editing.

CR Ambati: formal analysis and writing—original draft, review, and editing.
E Tycksen: formal analysis and writing—review and editing.
SJ Han: resources and writing—review and editing.
SE Bulun: resources and writing—review and editing.
N Putluri: formal analysis and writing—review and editing.
SW Biest: resources and writing—review and editing.
R Kommagani: conceptualization, data curation, formal analysis, funding acquisition, and writing—original draft, review, and editing.

### Conflict of Interest Statement

The authors declare that they have no conflict of interest.

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
