## [Reviewer comments · Life Science Alliance]

Life Science Alliance

Gut microbiota-derived short-chain fatty acids protect against the progression of endometriosis

Sangappa Chadchan, Pooja Popli, Chandrashekar Ambati, Eric Tycksen, Sang Han, Serdar Bulun, Nagireddy Putluri, Scott Biest, and Ramakrishna Kommagani

DOI: <https://doi.org/10.26508/lsa.202101224>

Corresponding author(s): Ramakrishna Kommagani, Washington University in St. Louis

Review Timeline:

Submission Date:	2021-09-02
Editorial Decision:	2021-09-08
Revision Received:	2021-09-20
Accepted:	2021-09-21

Transaction Report:

Please note that the manuscript was previously reviewed at another journal and the reports were taken into account in the decision-making process at *Life Science Alliance*. Since the original reviews are not subject to Life Science Alliance's transparent review process policy, the reports and author response cannot be published.

September 8, 2021

RE: Life Science Alliance Manuscript #LSA-2021-01224-T

Dr. Ramakrishna Kommagani
Washington University in St. Louis School of Medicine
Obstetrics and Gynecology
425 S. Euclid Avenue Campus Box 8064
BJC Institute of Health - 10th Floor, RM 10606
St. Louis 63132

Dear Dr. Kommagani,

Thank you for submitting your revised manuscript entitled "Gut microbiota-derived short-chain fatty acids protect against the progression of endometriosis". We would be happy to publish your paper in Life Science Alliance pending final revisions necessary to meet our formatting guidelines.

- please upload your main manuscript text as an editable doc file
- please upload your Tables in editable .doc or excel format
- please upload your main and supplementary figures as single files
- please add ORCID ID for the corresponding author-you should have received instructions on how to do so
- please add a Summary Blurb/Alternate Abstract in our system
- please add a Category for your manuscript in our system
- please add the Twitter handle of your host institute/organization as well as your own or/and one of the authors in our system
- please consult our manuscript preparation guidelines <https://www.life-science-alliance.org/manuscript-prep> and make sure your manuscript sections are in the correct order
- please add Author Contributions of all Authors in our system
- please add your main, supplementary figure, and table legends to the main manuscript text after the references section
- figure S4 is labeled incorrectly. Please correct
- please add callouts for Figures S4A-D, S9A-C, and S10A-C to your main manuscript text
- please add a Data Availability Statement with information to access the RNA-seq data

Figure check:

- missing scale bars in Figure 1F
- please add sizes next to the blots in Figure 6D and S6A
- scale bars in Figures 3G, S1F, S5A are not readable

LSA now encourages authors to provide a 30-60 second video where the study is briefly explained. We will use these videos on social media to promote the published paper and the presenting author. Corresponding or first-authors are welcome to submit the video. Please submit only one

video per manuscript. The video can be emailed to contact@life-science-alliance.org

A. FINAL FILES:

B. MANUSCRIPT ORGANIZATION AND FORMATTING:

****Reviews, decision letters, and point-by-point responses associated with peer-review at Life Science Alliance will be published online, alongside the manuscript. If you do want to opt out of**

having the reviewer reports and your point-by-point responses displayed, please let us know immediately.**

Sincerely,

September 21, 2021

RE: Life Science Alliance Manuscript #LSA-2021-01224-TR

Dr. Ramakrishna Kommagani
Washington University in St. Louis
Obstetrics and Gynecology
425 S. Euclid Avenue Campus Box 8064
BJC Institute of Health - 10th Floor, RM 10606
St. Louis 63132

Dear Dr. Kommagani,

Thank you for submitting your Research Article entitled "Gut microbiota-derived short-chain fatty acids protect against the progression of endometriosis". It is a pleasure to let you know that your manuscript is now accepted for publication in Life Science Alliance. Congratulations on this interesting work.

DISTRIBUTION OF MATERIALS:

Again, congratulations on a very nice paper. I hope you found the review process to be constructive and are pleased with how the manuscript was handled editorially. We look forward to future exciting submissions from your lab.

Sincerely,
